# 🖼️ Domain-RAG:

# Retrieval-Guided Compositional Image Generation for Cross-Domain Few-Shot Object Detection

Yu Li[1*]    Xingyu Qiu[1*]    Yuqian Fu[2*†]    Jie Chen[3]    Tianwen Qian[4]    Xu Zheng[2,5]
Danda Pani Paudel[2]    Yanwei Fu[1]    Xuanjing Huang[1]    Luc Van Gool[2]    Yu-Gang Jiang[1]

[1]Fudan University    [2]INSAIT, Sofia University "St. Kliment Ohridski"
[3]Fuzhou University    [4]East China Normal University    [5]HKUST(GZ)

## Abstract

Cross-Domain Few-Shot Object Detection (CD-FSOD) aims to detect novel objects with only a handful of labeled samples from previously unseen domains. While data augmentation and generative methods have shown promise in few-shot learning, their effectiveness for CD-FSOD remains unclear due to the need for both visual realism and domain alignment. Existing strategies, such as copy-paste augmentation and text-to-image generation, often fail to preserve the correct object category or produce backgrounds coherent with the target domain, making them non-trivial to apply directly to CD-FSOD. To address these challenges, we propose **Domain-RAG**, a training-free, retrieval-guided compositional image generation framework tailored for CD-FSOD. Domain-RAG consists of three stages: domain-aware background retrieval, domain-guided background generation, and foreground-background composition. Specifically, the input image is first decomposed into foreground and background regions. We then retrieve semantically and stylistically similar images to guide a generative model in synthesizing a new background, conditioned on both the original and retrieved contexts. Finally, the preserved foreground is composed with the newly generated domain-aligned background to form the generated image. Without requiring any additional supervision or training, Domain-RAG produces high-quality, domain-consistent samples across diverse tasks, including CD-FSOD, remote sensing FSOD, and camouflaged FSOD. Extensive experiments show consistent improvements over strong baselines and establish new state-of-the-art results. The source code and instructions are available at https://github.com/LiYu0524/Domain-RAG.

## 1 Introduction

Cross-Domain Few-Shot Object Detection (CD-FSOD) [10], an emerging task derived from cross-domain few-shot learning (CD-FSL) [13], aims to tackle few-shot object detection (FSOD) across different domains. Unlike conventional FSOD [20], which assumes source and target data share similar distributions, CD-FSOD considers more realistic scenarios with significant domain shifts, for example, transferring from natural images to industrial anomaly images, remote sensing imagery, or underwater environments. By simultaneously involving the challenges of few-shot learning and domain shift, CD-FSOD poses significant challenges for existing detectors.

---

*These authors have equal contributions.
†Corresponding author.

39th Conference on Neural Information Processing Systems (NeurIPS 2025).

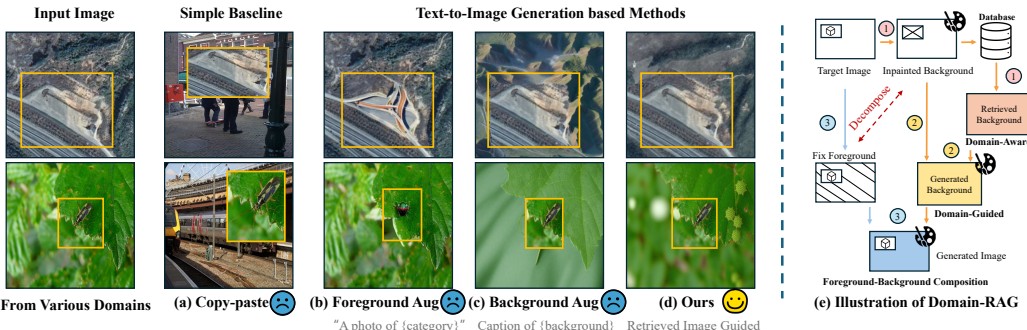

Figure 1: Given images from distinct novel domains, we compare generation results of baseline methods (a–c) and our approach (d), and illustrate the main pipeline of our Domain-RAG (e).

Due to the extreme scarcity of labeled data, e.g., as few as 1 or 5 annotated samples per category, a natural and intuitive solution is to leverage data augmentation to alleviate the data bottleneck. Although image augmentation and generation techniques have been extensively studied and shown effective in other few-shot learning tasks [61, 27, 29, 30], it remains unclear whether they can produce high-quality training samples for CD-FSOD. Different from the few-shot classification or in-domain FSOD, this setting requires not only accurate object annotation but also strong domain consistency, which has not been tackled in prior data augmentation-based few-shot learning methods.

To generate training images for CD-FSOD, the most straightforward approach is copy-paste (Fig. 1(a)). While easy to implement, such images often lack realism and domain coherence. A more advanced strategy is to build on recent generative models, particularly the trending text-to-image generation, such as SDXL [43], FLUX [21]. Most existing methods in this area focus on synthesizing foreground objects via text prompts, such as "a photo of category" (Fig. 1(b)). However, they might fail to preserve the object semantics when applied to novel categories and domains. Such a category shift is problematic for CD-FSOD, which has to tackle fine-grained objects and the domain gap. Other approaches generate diverse backgrounds (Fig. 1(c)), guided by text descriptions of the image. While this better preserves the foreground, purely textual descriptions often fall short of capturing precise domain characteristics and struggle to ensure semantic and visual consistency between foreground and background. These limitations motivate us to develop a new image generation framework capable of synthesizing visually coherent, domain-aligned training samples for CD-FSOD. Specifically, we aim to: ① preserve the original foreground object, ② generate diverse backgrounds that are both semantically and stylistically aligned with the query image and its domain, and ③ produce visually realistic images with valid annotations suitable for downstream detection training.

To that end, we propose **Domain-RAG**, a *retrieval-guided compositional image generation framework* built upon the principle of *fix the foreground, adapt the background*. Leveraging the nature of object detection, Domain-RAG begins by decomposing the target image into its foreground object and background, where the background is recovered by applying an inpainting model [49] to the object-masked region. Although simple in principle, this step is critical for preserving the original object and its annotations, laying the foundation for controllable compositional generation. The core challenge then lies in generating a new background that is semantically and stylistically compatible with the foreground. Inspired by the paradigm of retrieval-augmented generation (RAG) [24], we inject structured visual priors into the generative process to guide background synthesis. As illustrated in Fig. 1(e), Domain-RAG consists of the following three stages: **1) Domain-Aware Background Retrieval.** We introduce an image database (e.g., COCO [31]) containing diverse natural scenes, from which we retrieve candidate backgrounds that are semantically and stylistically similar to the inpainted background of the target image. Semantic similarity is computed using high-level visual features, while style similarity is measured via style-based descriptors [16]. **2) Domain-Guided Background Generation.** Rather than using the retrieved backgrounds directly, we feed them along with the target's inpainted background into a generative model to synthesize a new background that better reflects the visual characteristics of the target domain. To ensure compatibility with modern diffusion models, Redux [23] is applied to convert visual image cues into descriptive text prompts, enabling direct use of text-to-image generation models. **3) Foreground-Background Composition.** Finally, the preserved foreground is seamlessly composed onto the synthesized, domain-aligned background using a mask-guided generative model. The resulting image maintains the original object while embedding it in a realistic, domain-consistent context (Fig. 1(d)). The entire Domain-RAG pipeline is

*training-free* and can be directly integrated with existing detectors without any additional supervision or retraining, making it particularly suitable for low-shot scenarios such as 1-shot CD-FSOD.

We validate Domain-RAG on three various tasks that address few-shot object detection with domain shifts: CD-FSOD, remote sensing FSOD (RS-FSOD), and camouflaged FSOD. In all tasks, our method consistently improves a strong baseline by an average of +7.3, +1.1, and +2.1 mAP under the lowest-shot setting, achieving new state-of-the-art (SOTA) performance. These results demonstrate its broad applicability and effectiveness across diverse domains.

Our main contributions are as follows: 1) We propose Domain-RAG, a training-free, model-agnostic, retrieval-guided compositional image generation framework for boosting cross-domain few-shot object detection. 2) Domain-RAG enables image generation that preserves the original foreground while synthesizing domain-aligned backgrounds, guided by semantically and stylistically similar retrieved examples. 3) We achieve consistent performance improvements and new state-of-the-art results across a broad range of CD-FSOD, remote sensing FSOD, and camouflaged FSOD tasks.

## 2 Related Works

**Cross-Domain Few-Shot Tasks**. Few-shot learning across domains has been widely studied [13, 50, 8, 29, 63, 54, 11, 64], but most works focus only on classification. The more realistic task of cross-domain few-shot object detection (CD-FSOD) [10, 9], which involves both recognizing and localizing objects, remains underexplored. Recent methods like CD-ViTO [10] and ETS [42] address CD-FSOD. CD-ViTO introduces the task with a closed-source setting (COCO as the only source), while ETS uses a more practical open-source setting [9] and leverages data augmentation via pretrained GroundingDINO [33]. In this paper, we adopt the open-source setting and further improve augmentation using retrieval-guided compositional generation.

**FSOD Beyond Domains**. Beyond classic CD-FSOD tasks, many FSOD or detection problems also involve domain shifts, even if not explicitly labeled as cross-domain. Two notable examples are Remote Sensing FSOD (RS-FSOD) [34] and Camouflaged FSOD [39]. RS-FSOD uses remote sensing images, which differ from natural scenes in color, perspective, and resolution, creating a clear domain gap. Camouflaged FSOD involves detecting objects that blend into their backgrounds—like fish underwater or animals in the wild—posing challenges for generalization. We include both tasks to assess our method under diverse and difficult cross-domain scenarios.

**Data Augmentation**. Data augmentation is a key technique for the vision community. Traditional methods for object detection, like copy-paste [12], cropping, and color jittering [2], are simple but offer limited semantic variety. Recently, generative models—especially text-to-image models like ControlNet [55], SDXL [43], FLUX [21], and FLUX-Fill [22] have enabled more advanced augmentations. Methods such as X-Paste [59], Lin et al. [30], and Zhang et al. [57] generate new foregrounds to paste on diverse backgrounds, while others [40, 56, 55, 3] use text prompts to jointly create foregrounds and backgrounds. However, these methods typically rely on large amounts of in-domain data for training, which limits their adaptability to novel categories or unseen domains. In contrast, our Domain-RAG is training-free and leverages retrieved real-world images as visual priors to generate domain-consistent samples, making it well-suited for CD-FSOD.

**Retrieval-Augmented Generation in Vision**. First introduced in NLP [24], retrieval-augmented generation (RAG) enhances outputs by incorporating relevant retrieved content as external knowledge. Its strong performance has led to applications in vision tasks such as image captioning [45, 25], visual question answering [32, 15], and image generation [1, 38, 60], and pose estimation. However, current RAG-based image generation methods are aimed at open-ended synthesis and are not suited for object detection, particularly in cross-domain few-shot settings, where both domain alignment and object fidelity are crucial. To the best of our knowledge, we are the first to introduce a RAG-inspired, training-free image generation framework specifically designed for CD-FSOD.

## 3 Proposed Method

**Problem Setup**. The CD-FSOD task aims to adapt an object detector from a source domain $\mathcal{D}_S$ to a target domain $\mathcal{D}_T$, where the data distributions $\mathcal{P}_S$ and $\mathcal{P}_T$ differ. We use the few-shot setting, i.e., $N$-way $K$-shot protocol to evaluate detection results in $\mathcal{D}_T$. Specifically, a support set $\mathcal{S}^{N \times K} \subset \mathcal{D}_T$

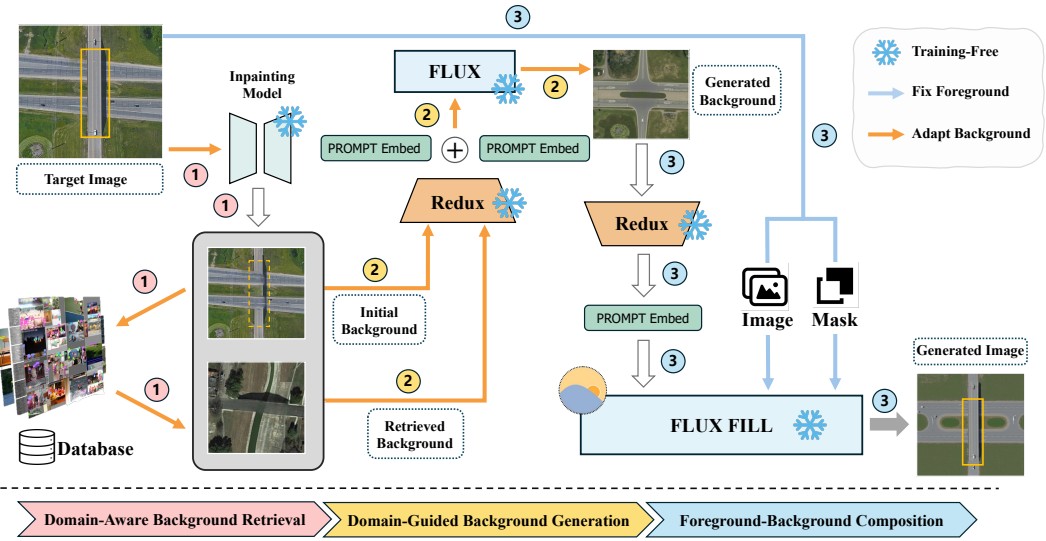

Figure 2: **Illustration of our Domain-RAG.** Built on our principle of "fix the foreground, adapt the background", we first decompose image and process it with three key modules: domain-aware background retrieval, domain-guided background generation, and foreground-background composition.

provides $K$ labeled examples per novel class, and a query set $\mathcal{Q}$ is used for evaluation. We use the open-source setting introduced in the 1st CD-FSOD Challenge [9], which allows foundation models pretrained on large-scale data, to explore the potential of foundation models in CD-FSOD. Particularly, instead of pretraining on $\mathcal{D}_S$, we directly finetune a pretrained detector (e.g., GroundingDINO [33]) on the support set $\mathcal{S}$ and evaluate the results on the query set $\mathcal{Q}$. To mitigate the limited size of $\mathcal{S}$, we augment each support instance with $n$ synthetic images, effectively expanding each class from $K$ to $K \times (n + 1)$ examples.

**Overview**. We propose Domain-RAG—a novel, training-free, retrieval-guided compositional generation framework that enhances support diversity by generating domain-aligned samples. To enable retrieval, we use COCO [31] as the database $\mathcal{D}_{base}$, serving as a gallery of candidate backgrounds. Following the core principle of "*fix the foreground, adapt the background*", Domain-RAG processes each support image $x \in \mathcal{S}$ by first decomposing it into foreground object(s) and background. As shown in Fig. 2, the framework then proceeds through three key stages: *1) domain-aware background retrieval* first obtains the inpainted background $b_{init}$ from $x$ and then retrieves $G$ candidate backgrounds $b_{re}$ from $\mathcal{D}_{base}$ that are semantically and stylistically similar. *2) domain-guided background generation* feeds each $\{b_{init}, b_{re}\}$ pair into a generative model to synthesize a new domain-aligned background $b_{dom}$. *3) foreground-background composition* finally produces $n$ new images $x^+$ by compositing the preserved foreground onto each $b_{dom}$ using a mask-guided generative model.

### 3.1 Domain-Aware Background Retrieval

We propose a two-stage retrieval strategy that combines CLIP's high-level semantic features with ResNet's low-level style descriptors to search an existing image database. The method retrieves images whose semantics and appearance are most similar to the target domain, providing background candidates that better match the target-domain distribution and thus enrich the support set $S$.

In practice, given a support image $x$, we remove the ground-truth bounding box with LaMa inpainting [49] to obtain a background without foreground $b_{init}$. We use the CLIP encoder to extract embeddings from the initial background $b_{init}$ and the database $\mathcal{D}_{base}$, which we refer to as $F_{bg}$ and $F_{base}$, respectively. We compute cosine similarity between the visual feature of the current background query $F_{bg}$ and the CLIP embedding of each sample in the database. The top $m$ most similar images are selected based on this similarity ranking, forming the candidate set $B_{<clip,m>}$, which contains $m$ images of the form $b_{clip}$. The subscript notation indicates that the set is constructed using CLIP vision encoder and contains $m$ elements.

Building on this step, we re-rank the $b_{clip}$ by extracting low-level style descriptors using shallow-layer ResNet features. For each background image $b_{clip}$ retrieved by CLIP, we extract its low-level feature

map $F$ using the early layers of a ResNet encoder. We further compute the per-channel mean $\mu_c$ and standard deviation $\sigma_c$ by averaging over the spatial dimensions of the feature map $F$. Concatenating the means and standard deviations over all channels yields a 128-D style vector as,

$$\mathbf{s}(b_{clip}) = [\mu_1, \ldots, \mu_C, \sigma_1, \ldots, \sigma_C] \in \mathbb{R}^{2C}. \tag{1}$$

For each retrieval candidate $b_{clip}$ in the set $B_{<clip,m>}$, we compute the style distance as the L2 norm between the style features of the original image $b_{init}$ and the candidate:

$$d = \|\mathbf{s}(b_{init}) - \mathbf{s}(b_{clip})\|_2. \tag{2}$$

Here, each $d$ corresponds to a candidate in $B_{<clip,\,m>}$, and we use the distance to rank and select the most stylistically similar backgrounds. We then re-rank the $m$ CLIP-retrieved candidates based on their style distances and retain the top $n$ images that are most similar in style. The resulting set of selected images is denoted as $B_{<re,\,n>}$. The images indexed by $B_{<re,\,n>}$ serve as style-matched references for the subsequent background generation stage.

## 3.2 Domain-Guided Background Generation

To fully leverage the retrieved images while keeping the generation process training-free, we adopt the Flux-Redux model[23] to encode each image into a prompt embedding. Given our domain-aware retrieval results $B_{<re,\,n>}$, let redux$(\cdot)$ denote FLUX-Redux encoder and FLUX$(\cdot)$ denote the FLUX generator. For each support image, we extract its clean background embedding $F_{bg} = \text{redux}(b_{init})$ and the embedding of the top retrieved image $b_{re} \in B_{<re,\,n>}$ as $F_{re} = \text{redux}(b_{re})$. We then fuse them as $F_{dom} = \lambda_1 F_{bg} + \lambda_2 F_{re}$, where $\lambda_1$ and $\lambda_2$ are hyper parameters.

Finally, the FLUX generator produces diverse background images at $1024 \times 1024$ resolution by applying a generative function FLUX to the domain embedding $F_{dom}$ i.e., $b_{dom} = \text{FLUX}(F_{dom})$. We sample this process $n$ times to generate a set of diverse images $\{b^{(1)}, b^{(2)}, \ldots, b^{(n)}\}$.

## 3.3 Foreground-Background Composition

Based on the diverse backgrounds generated in the previous stage, we aim to seamlessly integrate new backgrounds into the original images while preserving foreground pixels and maintaining the target-domain distribution. To achieve this, we employ Flux-Fill for outpainting. Specifically, for each corresponding support image $x$, we construct a binary mask $M \in \{0,1\}^{H \times W}$. The mask is computed based on the ground-truth bounding box bbox$(x)$ as:

$$M(p) = \begin{cases} 0, & \text{if } p \in \text{bbox}(x), \\ 1, & \text{otherwise.} \end{cases} \quad \text{for each } p \in \Omega_x, \tag{3}$$

where $\Omega_x$ denotes the spatial domain of image $x$, and $p$ indexes a pixel location. We then extract the prompt embedding $F_{gen}$ by $F_{gen} = \text{redux}(b_{dom})$ and feed $\{x, M, F_{gen}\}$ into Flux-Fill. To preserve foreground details, Flux-Fill encodes the input $x$ using a VAE and blends the encoded latent features with the initial noise. However, due to the VAE-based downsampling, it struggles to retain fine-grained structures such as small objects. To mitigate this issue, before generation, we denote the up-sampling method $s_{up}$ on each image as,

$$s_{up}(x) = \begin{cases} 0, & \text{if width}(x) > 1024 \text{ and height}(x) > 1024, \\ 1, & \text{otherwise.} \end{cases} \tag{4}$$

After generation, we denote a corresponding down-sampling method $s_{down}$ as,

$$s_{down}(x) = \begin{cases} 0, & \text{if } s_{up}(x) = 0, \\ 1, & \text{otherwise.} \end{cases} \tag{5}$$

The model then repaints only the masked regions, merging the style of $b_{dom}$ while keeping the foreground object's appearance and position unchanged. The final output of Domain-RAG, denoted $x^+$, is given by,

$$x^+ = s_{down}\left(\text{Flux-Fill}\left(s_{up}(x), s_{up}(M), F_{gen}\right)\right). \tag{6}$$

This completes the foreground-background composition, yielding an augmented support image with a domain-aligned background and unchanged foreground objects.

### 3.4 Applying Domain-RAG to CD-FSOD

In principle, our proposed **Domain-RAG** framework can be seamlessly integrated with any existing detector to enhance its performance in cross-domain scenarios. As a training-free, plug-and-play data augmentation module, Domain-RAG requires no modification to the detection architecture or training pipeline. Once the augmented support images are generated, the model is fine-tuned on the combination of the original support set $\mathcal{S}$ and the generated samples. At inference time, Domain-RAG is not involved; the detector is evaluated directly on the original query set $\mathcal{Q}$.

## 4 Experiments

**Setups**. We conduct experiments on three FSOD tasks with domain shifts: **1) CD-FSOD:** Following the CD-ViTO benchmark [10], we evaluate on six diverse target domains: ArTaxOr [6] (photorealistic), Clipart1k [17] (cartoon), DIOR [26] (aerial), DeepFish [46] (underwater), NEU-DET [47] (industrial), and UODD [18] (underwater). **2) Remote Sensing FSOD (RS-FSOD):** In addition to DIOR, we include NWPU VHR-10 [41], a popular remote sensing dataset for FSOD. **3) Camouflaged FSOD:** We also test on CAMO-FS [39], a recent dataset with 47 categories where objects are deliberately camouflaged into the background. For each task, we follow the standard dataset splits and evaluation protocols: 1/5/10-shot for CD-FSOD, 3/5/10/20-shot for RS-FSOD, and 1/2/3/5-shot for Camouflaged FSOD. Results are reported using mean Average Precision (mAP).

Table 1: **Main results (mAP) on the CD-FSOD benchmark** under the 1/5/10-shot setting. † marks methods implemented or reproduced by us. Best results are highlighted in pink.

| | Method | Backbone | ArTaxOr | Clipart1k | DIOR | DeepFish | NEU-DET | UODD | Average |
|---|---|---|---|---|---|---|---|---|---|
| 1-shot | Meta-RCNN [53] | ResNet50 | 2.8 | - | 7.8 | - | - | 3.6 | / |
| | TFA w/cos [51] | ResNet50 | 3.1 | - | 8.0 | - | - | 4.4 | / |
| | FSCE [48] | ResNet50 | 3.7 | - | 8.6 | - | - | 3.9 | / |
| | DeFRCN [44] | ResNet50 | 3.6 | - | 9.3 | - | - | 4.5 | / |
| | Distill-cdfsod [52] | ResNet50 | 5.1 | 7.6 | 10.5 | NaN | NaN | 5.9 | / |
| | ViTDeT-FT [28] | ViT-B/14 | 5.9 | 6.1 | 12.9 | 0.9 | 2.4 | 4.0 | 5.4 |
| | Detic [62] | ViT-L/14 | 0.6 | 11.4 | 0.1 | 0.9 | 0.0 | 0.0 | 2.2 |
| | Detic-FT [62] | ViT-L/14 | 3.2 | 15.1 | 4.1 | 9.0 | 3.8 | 4.2 | 6.6 |
| | DE-ViT [58] | ViT-L/14 | 0.4 | 0.5 | 2.7 | 0.4 | 0.4 | 1.5 | 1.0 |
| | CD-ViTO [10] | ViT-L/14 | 21.0 | 17.7 | 17.8 | 20.3 | 3.6 | 3.1 | 13.9 |
| | GroundingDINO† [33] | Swin-B | 26.3 | 55.3 | 14.8 | 36.4 | 9.3 | 15.9 | 26.3 |
| | ETS† [42] | Swin-B | 28.1 | 55.8 | 12.7 | 39.3 | 11.7 | 18.9 | 27.8 |
| | **Domain-RAG (Ours)** | Swin-B | **57.2** | **56.1** | **18.0** | 38.0 | **12.1** | **20.2** | **33.6** |
| 5-shot | Meta-RCNN [53] | ResNet50 | 8.5 | - | 17.7 | - | - | 8.8 | / |
| | TFA w/cos [51] | ResNet50 | 8.8 | - | 18.1 | - | - | 8.7 | / |
| | FSCE [48] | ResNet50 | 10.2 | - | 18.7 | - | - | 9.6 | / |
| | DeFRCN [44] | ResNet50 | 9.9 | - | 18.9 | - | - | 9.9 | / |
| | Distill-cdfsod [52] | ResNet50 | 12.5 | 23.3 | 19.1 | 15.5 | 16.0 | 12.2 | 16.4 |
| | ViTDeT-FT [28] | ViT-B/14 | 20.9 | 23.3 | 23.3 | 9.0 | 13.5 | 11.1 | 16.9 |
| | Detic [62] | ViT-L/14 | 0.6 | 11.4 | 0.1 | 0.9 | 0.0 | 0.0 | 2.2 |
| | Detic-FT [62] | ViT-L/14 | 8.7 | 20.2 | 12.1 | 14.3 | 14.1 | 10.4 | 13.3 |
| | DE-ViT [58] | ViT-L/14 | 10.1 | 5.5 | 7.8 | 2.5 | 1.5 | 3.1 | 5.1 |
| | CD-ViTO [10] | ViT-L/14 | 47.9 | 41.1 | 26.9 | 22.3 | 11.4 | 6.8 | 26.1 |
| | GroundingDINO† [33] | Swin-B | 68.4 | 57.6 | 29.6 | 41.6 | 19.7 | 25.6 | 40.4 |
| | ETS† [42] | Swin-B | 64.5 | 59.7 | 29.3 | 42.1 | 23.5 | 27.7 | 41.1 |
| | **Domain-RAG (Ours)** | Swin-B | **70.0** | **59.8** | **31.5** | **43.8** | **24.2** | 26.8 | **42.7** |
| 10-shot | Meta-RCNN [53] | ResNet50 | 14.0 | - | 20.6 | - | - | 11.2 | / |
| | TFA w/cos [51] | ResNet50 | 14.8 | - | 20.5 | - | - | 11.8 | / |
| | FSCE [48] | ResNet50 | 15.9 | - | 21.9 | - | - | 12.0 | / |
| | DeFRCN [44] | ResNet50 | 15.5 | - | 22.9 | - | - | 12.1 | / |
| | Distill-cdfsod [52] | ResNet50 | 18.1 | 27.3 | 26.5 | 15.5 | 21.1 | 14.5 | 20.5 |
| | ViTDeT-FT [28] | ViT-B/14 | 23.4 | 25.6 | 29.4 | 6.5 | 15.8 | 15.6 | 19.4 |
| | Detic [62] | ViT-L/14 | 0.6 | 11.4 | 0.1 | 0.9 | 0.0 | 0.0 | 2.2 |
| | Detic-FT [62] | ViT-L/14 | 12.0 | 22.3 | 15.4 | 17.9 | 16.8 | 14.4 | 16.5 |
| | DE-ViT [58] | ViT-L/14 | 9.2 | 11.0 | 8.4 | 2.1 | 1.8 | 3.1 | 5.9 |
| | CD-ViTO [10] | ViT-L/14 | 60.5 | 44.3 | 30.8 | 22.3 | 12.8 | 7.0 | 29.6 |
| | GroundingDINO† [33] | Swin-B | 73.0 | 58.6 | 37.2 | 38.5 | 25.5 | 30.3 | 43.9 |
| | ETS† [42] | Swin-B | 70.6 | 60.8 | 37.5 | **42.8** | 26.1 | 28.3 | 44.4 |
| | **Domain-RAG (Ours)** | Swin-B | **73.4** | **61.1** | **39.0** | 41.3 | **26.3** | **31.2** | **45.4** |

132

**Implementation Details.** We use pretrained GroundingDINO [33] with Swin-Transformer [35] Base (Swin-B) as backbone as our baseline. For the retrieval stage, the hyper parameters are set to $m = 100$, $n = 5$, throughout all experiments. For the background generation stage, fusion hyper parameters $\lambda_1$ and $\lambda_2$ are set to 1.0 and 0.8 respectively. We fine-tune the model for 30 epochs by default, but reduce to 5 for faster-converging datasets like Clipart1k and DeepFish. We use AdamW [36] with learning rate and weight decay set to $1 \times 10^{-4}$, and we scale the backbone's learning rate by 0.1. All experiments are run on four Tesla V100 GPUs or eight 5880 Ada GPUs, or a single A800 GPU. Further details are in the Appendix.

## 4.1 Main Comparison Results

**CD-FSOD Results**. Tab. 1 summarizes the main comparison results on CD-FSOD under 1/5/10 shots across six novel targets. Particularly, we include several competitors: Meta-RCNN [53], TFA w/cos [51], FSCE [48], DeFRCN [44], Distill-cdfsod [52], ViTDeT-FT [28], Detic/Detic-FT [62], DE-ViT [58] as reported in CD-ViTO [10]. In addition, we also report the results of fine-tuned GroundingDINO [33], ETS [42] to compare with our Domain-RAG. Note that both ETS and Domain-RAG build on GroundingDINO but with different augmentation strategies.

We highlight that Domain-RAG consistently outperforms existing competitors across most target domains, achieving new state-of-the-art (SOTA) results. Compared to the GroundingDINO baseline, our method improves mAP by 7.3, 2.3, and 1.5 points under the 1, 5, and 10 shots, respectively. These results not only show the effectiveness of Domain-RAG, but also reveal its superiority over other proposed augmentation strategies such as ETS. Beyond the average gains, we also notice: 1) *Significant gains on ArTaxOr.* Domain-RAG achieves a 117.5% relative improvement in the 1-shot setting. We attribute this to the strong semantic and visual compatibility between ArTaxOr and the retrieved COCO-style backgrounds, where ArTaxOr features the fine-grained foreground but with a relatively close visual domain to COCO regarding background. 2) *Robustness under low-shot settings.* The largest gains are observed in the 1-shot scenario, which is the most challenging FSOD scenario. This shows our benefits under severe data scarcity. 3) *Strong generalization to severe domain shift.* On NEU-DET, an industrial defect detection dataset characterized by uncommon objects and background styles, Domain-RAG consistently improves all shot settings, demonstrating its capability to handle the most challenging cross-domain FSOD cases.

**RS-FSOD Results.** Tab. 2 summarizes the results on the NWPU VHR-10 remote sensing dataset [41] under the 3/5/10/20-shot settings. The dataset is divided into 7 base classes and 3 novel classes. The table is split into two parts. In the *upper part*, we follow the standard RS-FSOD protocol: models are first trained on the base classes and then fine-tuned and evaluated on the novel classes. Under this setting, the base classes contain a sufficient number of annotated samples, and we apply our augmentation strategy on top of the previous state-of-the-art method SEA-FSDet [34], and report the mean Average Precision (mAP) over the 3 novel classes. In the *lower part*, we explore the dataset in a CD-FSOD setting, where the pretrained model is directly fine-tuned on all 10 classes (both base and novel), each with only a few labeled samples. To ensure comparability with the upper setting, the reported mAP here reflects performance exclusively on the three novel categories.

Table 2: **Main results (mAP) on the NWPU VHR-10 benchmark** under the 3/5/10/20-shot settings. The upper part follows the standard **RS-FSOD** problem setup, while the lower part adapts **CD-FSOD** setup, with the best results highlighted in pink. † means results are produced by us.

| Method | Training Setting | Backbone | 3-shot | 5-shot | 10-shot | 20-shot | Average |
|---|---|---|---|---|---|---|---|
| Meta-RCNN [53] | RS-FSOD | ResNet-50 | 20.51 | 21.77 | 26.98 | 28.24 | 24.38 |
| FsDetView [19] | RS-FSOD | ResNet-50 | 24.56 | 29.55 | 31.77 | 32.73 | 29.65 |
| TFA w/cos [51] | RS-FSOD | ResNet-50 | 16.17 | 20.49 | 21.22 | 21.57 | 19.86 |
| P-CNN [4] | RS-FSOD | ResNet-50 | 41.80 | 49.17 | 63.29 | 66.83 | 55.27 |
| FSOD [7] | RS-FSOD | ResNet-50 | 10.95 | 15.13 | 16.23 | 17.11 | 14.86 |
| FSCE [48] | RS-FSOD | ResNet-50 | 41.63 | 48.80 | 59.97 | 79.60 | 57.50 |
| ICPE [37] | RS-FSOD | ResNet-50 | 6.10 | 9.10 | 12.00 | 12.20 | 9.85 |
| VFA [14] | RS-FSOD | ResNet-50 | 13.14 | 15.08 | 13.89 | 20.18 | 15.57 |
| SAE-FSDet [34] | RS-FSOD | ResNet-50 | 57.96 | 59.40 | 71.02 | **85.08** | 68.36 |
| **Domain-RAG (Ours)** | RS-FSOD | ResNet-50 | **59.99** | **65.78** | **72.87** | 84.05 | **70.67** |
| GroundingDINO† [33] | CD-FSOD | Swin-B | 57.1 | 61.3 | 65.1 | 69.5 | 63.3 |
| **Domain-RAG (Ours)** | CD-FSOD | Swin-B | **58.2** | **62.1** | **66.6** | **69.7** | **64.2** |

Notably, from the upper standard RS-FSOD results, we observe the following findings: 1) Our Domain-RAG achieves the best result via improving the strong SEA-FSDet, achieving 2.31 mAP improvement across all shots on average. This indicates that our plug-and-play augmented method is compatible with existing methods. 2) Minor decrease is observed for 20-shot, from 85.08 to 84.05. We speculate that it is due to the base training being sufficient. Further augmentation in this regime may lead to overfitting on synthetic data patterns rather than benefiting novel-class generalization. From the lower part of the CD-FSOD setting results, we highlight that our method again improves the strong GroundingDINO baseline, indicating its effectiveness.

**Camouflaged FSOD Results.** Tab. 3 presents the results on the CAMO-FS [39] under 1/2/3/5 shots. All categories in this dataset are treated as novel classes and are further split into a support set and a query set, naturally aligning with the formulation of CD-FSOD. The first two rows in the table report the results of "FS-CDIS-ITL" and "FS-CDIS-IMS", two methods developed from the original CAMO-FS paper. Below that, we include our reproduced baseline using GroundingDINO as the detector, along with the results of our proposed Domain-RAG method built on top of GroundingDINO.

As shown by the results, the large-scale pretrained model, i.e., GroundingDINO, brings a substantial performance boost to this task, improving results from around 7 to over 65 mAP.

We believe this remarkable advancement will advance the frontier of this field. Moreover, the performance gains introduced by our proposed method over the GroundingDINO baseline remain consistently clear across all shot settings. The consistent success across CD-FSOD, RS-FSOD, and camouflaged FSOD, covering eight challenging and diverse domains, demonstrates that our method serves as a general and effective solution for addressing the gap issue in few-shot object detection.

Table 3: **Main results (mAP) on the Camouflage FSOD** under the 1/2/3/5-shot settings. † means the results are produced by us, the best results are highlighted in pink.

| Method | Backbone | 1-shot | 2-shot | 3-shot | 5-shot | Average |
|---|---|---|---|---|---|---|
| FS-CDIS-ITL [39] | ResNet-101 | 4.0 | 7.3 | 7.5 | 9.8 | 7.1 |
| FS-CDIS-IMS [39] | ResNet-101 | 4.5 | 7.0 | 7.6 | 10.4 | 7.4 |
| GroundingDINO† [33] | Swin-B | 63.4 | 66.8 | 67.1 | 69.1 | 66.6 |
| **Domain-RAG (Ours)** | Swin-B | **65.5** | **67.7** | **68.3** | **70.3** | **68.0** |

## 4.2 Comparison with Other Augmentation Methods

To assess the effectiveness of our Domain-RAG framework, we compare it with several strong baselines that are designed for augmenting data for CD-FSOD. Specifically, 1) "Copy-Paste" directly overlays foreground objects onto random COCO backgrounds without considering semantic relevance or compositional integrity. 2) "Foreground Augmentation" attempts to diversify object appearances by inpainting new foregrounds after object removal. This is done by using the category label of each bounding box as a text prompt and applying SDXL-inpaint to generate a new foreground after removing the original object. 3) "Background Augmentation", which we use InstructBLIP [5] to caption the remaining background, and guide SDXL to generate a new background based on this caption after removing the foreground from a target image. To ensure fair comparison, all the augmentation methods use $G = 5$. Comparison results are summarized in Tab. 4. The results are reported on CD-FSOD under the 1-shot setting.

Table 4: Comparison of augmentation methods (mAP) on the CD-FSOD benchmark under 1-shot.

| Method | ArTaxOr | Clipart | DIOR | DeepFish | NEU-DET | UODD | Average |
|---|---|---|---|---|---|---|---|
| GroundingDINO | 26.3 | 55.3 | 14.8 | 36.4 | 9.3 | 15.9 | 26.3 |
| Copy-Paste | 38.8 | 55.0 | 15.0 | 36.4 | 8.4 | 14.2 | 27.9 |
| Foreground Augmentation | 32.4 | 56.1 | 13.9 | **41.4** | 9.6 | 14.9 | 28.1 |
| Background Augmentation | 52.3 | 53.7 | 16.9 | 34.2 | 8.9 | 10.8 | 29.5 |
| **Domain-RAG (Ours)** | **57.2** | **56.1** | **18.0** | 38.0 | **12.1** | **20.2** | **33.6** |

We observe that: 1) copy-paste methods can work reasonably well on relatively simple datasets such as ArTaxOr. However, due to a lack of semantic consistency and domain alignment, they tend to fail on most target domains. 2) Foreground-augmentation baseline performs well when the foreground is visually simple and isolated, for example, in datasets like DeepFish, where only a single object is present. However, due to the potential semantic shift issue, it failed on more complex datasets such as DIOR and UODD. 3) Background-augmentation baseline also suffers in CD-FSOD, often failing on datasets with distinctive domain characteristics, such as NEU-DET. 4) In contrast, our method consistently improves upon the baseline across all datasets, demonstrating its robustness and effectively addressing the limitations of prior approaches.

## 4.3 More Analysis

**Ablation Study on Proposed Modules.** To evaluate each module's effectiveness, we conduct ablation studies by removing or replacing it with naive alternatives. As a typical challenging case, the NEU-DET under a 1-shot setting is demonstrated as an example. Results are shown in Fig. 5 (a). Specifically, 1) the grey bar marks the "baseline", i.e., vanilla fine-tuned GroundingDINO. 2) The pink bar ("w/o background retrieval") disables the domain-aware retrieval module and replaces the backgrounds with random COCO images while keeping the rest of the pipeline unchanged. 3) The yellow bar ("w/o background generation") skips the domain-guided background generation and directly performs the foreground-background composition with the raw retrieved images from COCO. 4) The blue bar ("copy-paste as compositional") removes the last foreground–background composition stage and simply pastes the foreground onto the domain-aligned generated background. 5) The last colorful bar represents our full Domain-RAG.

Results show that our full model outperforms all ablated variants, achieving the best overall performance. Furthermore, we observe the following: 1) By comparing our method with the pink bar, we verify that the domain-aware background retrieval stage provides backgrounds that are better aligned with the target domain. 2) The comparison between the gray and yellow bars indicates that simply augmenting backgrounds using COCO images offers limited benefits. In contrast, the domain-guided background generation stage significantly improves performance by producing backgrounds that are both semantically and stylistically aligned, as evidenced by the gap between the yellow and final colorful bars. 3) The performance drop seen with the blue bar underscores the importance of the foreground-background composition stage, which enables seamless integration of foreground objects into the generated backgrounds. Together, these observations confirm that each component of Domain-RAG is both indispensable and complementary for achieving robust CD-FSOD performance.

**The Construction of RAG Database** In the defined (closed-source) CD-FSOD setting as proposed in CD-ViTO [10], COCO serves as the only single-source dataset for training, while other datasets (ArTaxOr, Clipart1k, DIOR, DeepFish, NEU-DET, UODD) are treated as unseen targets. Using COCO as the RAG database brings two key advantages: (1) It does not introduce any extra data beyond the default setting, ensuring the fairness of comparison; (2) COCO provides diverse and general-domain backgrounds that better cover novel domain scenarios.

Furthermore, we conducted additional experiments using different database options, including COCO with reduced category numbers, NEU-DET (non-general-domain), and miniImageNet.

Table 5: Effect of different database choices on Domain-RAG performance.

| DataBase | ArTaxOr | Clipart1k | DIOR | FISH | NEU-DET | UODD | Avg |
|---|---|---|---|---|---|---|---|
| Base (GroundingDINO) | 26.3 | 55.3 | 14.8 | 36.4 | 9.3 | 15.9 | 26.3 |
| COCO-1class | 50.1 | 55.0 | 15.7 | 36.6 | 12.0 | 16.0 | 30.9 |
| COCO-5classes | 51.0 | 55.1 | 16.6 | 36.7 | 11.9 | 17.5 | 31.5 |
| COCO-20classes | 53.0 | 56.2 | 16.2 | 37.0 | 12.2 | 18.9 | 32.3 |
| **COCO-80classes (Ours)** | **57.2** | **56.1** | **18.0** | **38.0** | 12.1 | **20.2** | **33.6** |
| NEU-DET | 49.8 | 55.2 | 16.4 | 37.0 | 12.0 | 16.1 | 31.1 |
| miniImageNet | 55.6 | 53.2 | 15.6 | 38.0 | **14.0** | 16.2 | 32.1 |

From the results summarized in Table 5, we observe that: (1) broader category coverage consistently improves performance; (2) general-domain databases such as COCO outperform specific-domain

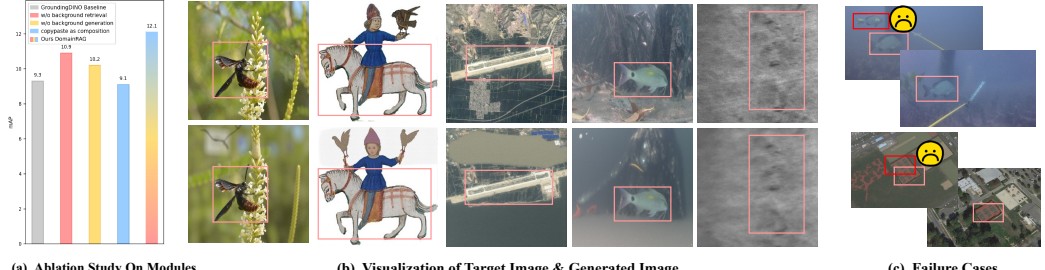

| (a) Ablation Study On Modules | (b) Visualization of Target Image & Generated Image | (c) Failure Cases |

Figure 3: (a) Ablation study on modules, results reported on NEU-DET, 1 shot. (b) Visualization of target image (top row) and generated image (second row). (c) Failure Cases.

ones like NEU-DET; and (3) although miniImageNet can serve as an alternative database, it performs slightly worse than COCO due to its larger foreground regions and less diverse backgrounds. These findings demonstrate that our Domain-RAG consistently enhances the base GroundingDINO across all benchmarks, validating the robustness and effectiveness of our approach.

**Visualization of Generation Images.** To provide a more intuitive illustration of our method's effectiveness, we present qualitative results in Fig. 5 (b). Each example shows the original target image from a different domain in the first row and the corresponding generated image in the second row, with annotated object bounding boxes. From the results, we observe the following: 1) Our method effectively preserves the foreground object without introducing noticeable changes, even in challenging domains such as remote sensing, underwater, and industrial defect scenarios. 2) The generated images successfully introduce new backgrounds while maintaining overall semantic coherence and visual consistency with the original domain. Also, the outputs appear natural and realistic. These two observations align well with our goals and further validate the effectiveness of the proposed Domain-RAG framework.

**Failure Cases and Limitations.** We further examine the quality of the generated images and observe that, in a few cases, our model exhibits foreground information leakage. Parts of the foreground object are unintentionally regenerated within the background, as illustrated in the area highlighted with red boxes of Fig. 5 (c). Since these regenerated foregrounds are not explicitly controlled and lack corresponding annotations, they may introduce noise into model fine-tuning and potentially harm detection performance.

Additional results, including ablation studies of our proposed modules on other targets, more detailed analyses, and extended visualizations, are provided in the Appendix.

## 5 Conclusion

In this paper, we investigate few-shot object detection (FSOD) across domains—a more realistic yet significantly more challenging scenario than conventional FSOD. We focus on three representative tasks: cross-domain FSOD (CD-FSOD), remote sensing FSOD (RS-FSOD), and camouflaged FSOD. To improve performance under these settings, we propose **Domain-RAG**, a training-free compositional image generation framework designed to produce domain-aligned and detection-friendly samples. Unlike existing text-to-image generation approaches that rely solely on textual prompts, Domain-RAG retrieves semantically and stylistically similar images as structured priors to guide the generation process. To the best of our knowledge, this is the first application of retrieval-augmented generation to cross-domain object detection, particularly in a training-free way suitable for low-shot scenarios. Domain-RAG achieves new state-of-the-art results across all three tasks, demonstrating its generalization ability and opening new directions for training-free data synthesis.

## 6 Acknowledgment

This work was supported by the Science and Technology Commission of Shanghai Municipality (No. 24511103100).The authors gratefully thank the organization for their support and resources.

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
