# OpenReview forum: "Domain-RAG: Retrieval-Guided Compositional Image Generation for Cross-Domain Few-Shot Object Detection"
_NeurIPS.cc/2025/Conference — NeurIPS 2025 poster_

### Official Review · Reviewer_Dhsv · 2025-06-20

**Clarity:** 2
**Significance:** 2
**Originality:** 3
**Rating:** 4
**Confidence:** 2

**Summary:**

This paper aims to tackle the task of Cross-Domain Few-Shot Object Detection (CD-FSOD), which involves detecting new-class objects with a limited number of labels in unseen domains. The authors find that common data augmentation methods, such as copy-paste, used in previous works, cannot generate realistic samples for this task. They then propose a comprehensive data framework, named Domain-RAG, with a training-free nature, to augment data for CD-FSOD. This framework consists of three stages. It decouples the image into foreground and background, and retrieves similar images in a dataset, based on which it further simulates new backgrounds. Finally, the proposed method combines the objects and simulated background into a new image, serving for the model training. Experiments on three FSOC datasets show that the proposed method can achieve noticeable performance gains.

**Questions:**

- Considering the usage of several generative models, how long will it take to generate one image? This is critical for dataset scaling.
- It is suggested to report the results of baseline GroundingDINO with and without fine-tuning.
- How about the efficiency of Domain-RAG in comparison with other related works?
- The proposed method uses a stronger backbone. How will the performance be using consistent backbones with other counterparts?
- Will the proposed method improve the generic few-shot object detection without other domains? I did not find the specific designs for the studied cross-domain setting. Please give some comparisons with state-of-the-art methods.

**Ethical Concerns:**

["NO or VERY MINOR ethics concerns only"]

**Final Justification:**

Thanks for the detailed response, which has addressed my concerns. Hence, I am happy to keep my original positive score.

**Limitations:**

Yes

**Quality:**

3

**Strengths And Weaknesses:**

Strength:
- The paper is easy to follow
- The proposed method can synthesize high-quality samples with diverse backgrounds, which is useful for some design perception tasks. The quality of the generated samples in the Appendix seems satisfactory.
- The method is training-free, fully using existing foundational models.
- The experiment performance is state-of-the-art; The ablation of different types of generation designs is interesting.

Weakness
- The proposed method potentially has general usage instead of being tailored for this specific task (CD-FSOD). The studied scope is limited.
- Some experiment details need to be clarified. Besides, due to the extra usage of data and models, the comparison may be unfair
- The model efficiency, such as parameters and inference speed, of the proposed and compared methods should be reported

---

> ### Author Rebuttal · Authors · 2025-07-31
>
> Thank you very much for the positive and constructive feedback! We really appreciate that you consider our work “easy to follow”, “training-free”, “can synthesize high-quality samples”, and delivering “state-of-the-art results and interesting ablations”. Below, we address the concerns raised.
>
> # W1&Q5:  Potential General Usage Instead of being Tailored for CD-FSOD.
> Thanks. We appreciate this constructive suggestion. However, we would also like to highlight that:
>
> 1) We focus on CD-FSOD mainly due to its unique and challenging characteristics: significant domain shift and limited training examples.  These two challenges make almost all the prior image generation can’t work well.  In contrast, our method perfectly addresses their limitations with our training-free, retrieval-based augmentation strategies, showing a clear performance boost over other methods. That said, if this is not the case with CD-FSOD, we most likely could propose new solutions with better solutions, for example, finetuning when the examples are not few-shot.
>
> 2) Besides, in addition to the CD-FSOD it alone, in our paper, we also considered and conducted experiments on other FSOD tasks: in Tab.2, we studied the standard Remote-Sensing FSOD (which falls within general FSOD instead of CD-FSOD), and the adapted Cross-Domain Remote-Sensing FSOD; in Tab.3, we studied the CamouflageFSOD. These broader experiments demonstrate our effectiveness beyond CD-FSOD.
>
> For more broader scope, e.g., general object detection, we agree our method has potential usage but also we think the current work might not the optimal form, so we will take the suggestion and explore it as our future work.
>
> # W2: Experiment Details and Fairness Due to Data Usage and Models.
> Thanks. For the experiment details, in addition to Setups and Implementation Details provided in Sec.4 (L204-L220), we also provide more details in Sec.A.1(Appendix), including more details on the proposed method (Sec.A.1.1), more details on training (Sec.A.1.2), and more ablation studies (Sec.A.1.3).
>
>
> For the Fairness issue, on the data side, we clarify that: though we use COCO as a database to augment new images, COCO itself is defined as the source in the default CD-FSOD setting, and we don’t break this formulation. It fundamentally should be regarded as different ways of using the source data. Additionally, 1) the comparison between ours (33.6, Tab.1) and other competitors, e.g., ETS (27.8, Tab.1),  Copy-Paste (27.9, Tab.4), Foreground Aug. (28.1, Tab.4), and Background Aug. (29.5, Tab.4) is fair: we share the exact same baseline and training configuration, and all of them are data augmentation-based approaches. For example, ETS proposes a mixture of 6 various augmentation strategies during training;  2) the ablation experiments on Fig.3(a) and Tab.5 (Appendix) all use COCO as DB while without applying the complete Domain-RAG method. We believe these results confirm that the improvements stem from our pipeline design rather than merely the introduction of synthetic data.
>
> On the model side, since we aim to build a training-free augmentation, the introduction of extra generation models, e.g., FLUX, is unavoidable. However, we highlight: 1) As stated before, the baseline (GroundingDINO) is the same as others; 2) All the extra introduced models, e.g., FLUX, are only used for the generation; during the inference time, only the single base GroundingDINO is adopted, which is also the same as others.
>
> # W3&Q1&Q3: Comparison of Model Efficiency, such as Parameters and Inference Speed.
>
> Thanks, our method builds on GroundingDINO without modifying any model architecture or training/inference process on the target support sets. Therefore, both parameter count (233.00 M) and inference speed (~0.22 seconds per image) are the same as those of competitors that are built on the same GroundingDINO, e.g., GroundingDINO itself, EST, and our Domain-RAG.
>
> In addition, upon request, we further provide the computation time needed for our image generation below. We emphasize that: 1) the generation indeed takes time, while it is a long-standing issue in the whole generation community. 2) The generation period introduces no additional cost during inference. 3) The current bottleneck of CD-FSOD lies in the model performance, due to its huge challenges, while our method provides practical and effective solutions for enhancing existing detectors for CD-FSOD.
>
> These together make our method still scalable, especially under low-shot cases, where only a few query images are needed to be augmented.
>
>
> | Method              | Background Retrieval | Background Generation | Image Composition | Total Time (s/img) | Avg. mAP (1-shot) |
> | :------------------ | :------------------: | :-------------------: | :---------------: | :----------------: | :---------------: |
> | Baseline            |          ✗           |           ✗           |         ✗         |         ✗          |       26.3        |
> | Copy-Paste          |        ✓ (0.1)        |           ✗           |      ✓ (0.1)      |       ~0.2         |       27.9        |
> | Foreground Aug      |        ✓ (0.1)        |           ✗           |      ✓ (8.0)      |       ~8.1         |       28.1        |
> | Background Aug      |          ✗           |        ✓ (3.0)         |      ✓ (15.0)     |       ~18.0        |       29.5        |
> | Domain-RAG (Ours) |     ✓ (0.14)      |     ✓ (27.0)       |    ✓ (28.0)   |    ~55.14      |     **33.6**      |
>
> # Q2: Grounding-DINO with/without Fine-tuning.
> Thanks, we provide a comparison of GroundingDINO under different training strategies: zero-shot, 1-shot fine-tuning, and with Domain-RAG.
>
> We notice that:
> -   Zero-shot GroundingDINO shows limited generalization, especially on challenging datasets (e.g., ArTaxOr, NEU-DET).
> -   Fine-tuning with 1-shot improves results moderately.
> -   Domain-RAG with fine-tuned GroundingDINO achieves consistent gains across all datasets, with a +30.9 mAP increase on ArTaxOr and +4.3 mAP on UODD compared to 1-shot GroundingDINO.
>
> This demonstrates that GroundingDINO is a strong baseline and Domain-RAG helps improve its generalization in few-data, cross-domain scenarios further.
>
> | Method                          | ArTaxOr | Clipart1k | DIOR | DeepFish | NEU-DET | UODD | **Avg.** |
> | :------------------------------ | ------: | --------: | ---: | -------: | ------: | ---: | -------: |
> | GroundingDINO (ZSL)             |    11.3 |      54.1 | 15.6 |     35.3 |     3.1 | 13.9 |     22.2 |
> | GroundingDINO (1-shot) (Tab.1)  |    26.3 |      55.3 | 14.8 |     36.4 |     9.3 | 15.9 |     26.3 |
> | **Domain-RAG (Ours) (Tab.1)**   |  **57.2**|    **56.1**|**18.0**|   **38.0**|  **12.1**|**20.2**|   **33.6** |
>
> # Q4:  A Stronger Backbone Used.
> As mentioned in W2, we conducted controlled experiments using the same backbone and training setup as competitors (e.g., ETS) to ensure a fair comparison.
>
> Domain-RAG consistently outperforms these baselines, showing that the performance gain comes from our augmentation method rather than differences in model capacity. To further demonstrate the generality of Domain-RAG, we applied it to DEViT-FT and CD-ViTO. As shown below, both “DEViT-FT +  Domain-RAG” and “CD-ViTO + Domain-RAG” yield consistent improvements across all datasets.
>
> | Method                | ArTaxOr | Clipart1k | DIOR | DeepFish | NEU-DET | UODD |
> | :-------------------- | ------: | --------: | ---: | -------: | ------: | ---: |
> | DeViT-FT              |    10.5 |      13.0 | 14.7 |     19.3 |     0.6 |  2.4 |
> | DeViT-FT + Ours   |    10.4 |    17.0 | 15.7 |     19.1 |     2.4 |  3.0 |
> | CD-ViTO               |    21.0 |      17.7 | 11.7 |     20.3 |     3.6 |  3.1 |
> | **CD-ViTO + Ours**    |    **22.5** |    **20.1** | 15.7 |    **22.4** |     3.6 |  3.3 |
>
> These results confirm that Domain-RAG improves detection performance across different model architectures, supporting its applicability as a general-purpose augmentation strategy. We will include these in the revised version.
>
>
> We hope that our response adequately addresses your concerns. We are happy to provide more information during the discussion phase.

---

> > ### Comment · Reviewer_Dhsv · 2025-08-02
> >
> > Thanks for the detailed response, which has addressed my concerns. Hence, I am happy to keep my original positive score.

---

> > > ### Author Response · Authors · 2025-08-04
> > > **Thanks for your response!**
> > >
> > > Dear Reviewer Dhsv,
> > >
> > > Thank you very much for taking the time to respond during the discussion phase. We’re glad to hear that our rebuttal has addressed your concerns, and we truly appreciate your constructive suggestions. We'll revise our paper according to your questions and suggestions.
> > >
> > > If there are any remaining concerns or aspects you believe we could further improve, we would be very happy to address them. Please let us know if there’s anything else we can clarify or do to better support a higher recommendation.
> > >
> > > Thank you again for your time and engagement.
> > >
> > > Best regards,
> > > Authors of Submission#4884

---

### Official Review · Reviewer_vndr · 2025-06-30

**Clarity:** 3
**Significance:** 2
**Originality:** 2
**Rating:** 4
**Confidence:** 4

**Summary:**

This paper introduces Domain-RAG, a retrieval-guided compositional image generation framework designed to enhance performance in Cross-Domain Few-Shot Object Detection (CD-FSOD). Domain-RAG follows a training-free three-stage pipeline: (1) domain-aware background retrieval, (2) domain-guided background generation, and (3) foreground-background composition. The method leverages visual retrieval and generative models to synthesize domain-aligned backgrounds that preserve original foreground objects. Extensive experiments demonstrate that Domain-RAG achieves consistent improvements over existing methods, setting new state-of-the-art results in various few-shot detection scenarios.

**Questions:**

1. In scenarios where the generative models unintentionally leak foreground information into the background (as noted in failure cases), what additional controls or safeguards could mitigate this issue?
2. How well does this method adapt to zero-shot scenarios?
3. How are the hyperparameters $\lambda_1$ccand $\lambda_2$ selected, and do they significantly affect performance?
4. The caption of Figure 3 contains a typo. It should be "(a)" instead of "a)".
5. How does the number of augmented samples per category affect the performance? How is an appropriate number of synthesized samples determined?

**Ethical Concerns:**

["NO or VERY MINOR ethics concerns only"]

**Limitations:**

See Questions.

**Paper Formatting Concerns:**

None.

**Quality:**

3

**Strengths And Weaknesses:**

Strengths:
1. Domain-RAG creatively combines retrieval-augmented generation techniques with compositional image synthesis, effectively addressing the critical challenge of domain consistency in CD-FSOD tasks.
2. The use of retrieval-based visual priors effectively guides the generative process, significantly improving background realism and coherence compared to purely generative methods.
3. Clear state-of-the-art results are achieved across all evaluated few-shot settings, demonstrating robust performance gains compared to strong baselines.

Weaknesses:
1. The paper primarily presents empirical findings without rigorous theoretical analysis or deeper conceptual insights, which could limit its novelty for a NeurIPS audience.
2. The performance significantly depends on the quality and domain relevance of the retrieval database (e.g., COCO), which might limit generalization to datasets drastically different from standard benchmarks.

---

> ### Author Rebuttal · Authors · 2025-07-31
>
> Thank you very much for the positive and constructive feedback! We really appreciate that you consider our work “creative”, “effective”, “significantly improving background realism and coherence”, and with “clear state-of-the-art results” . Below we address the concerns raised.
>
> # W1: Primarily on Empirical Findings without Rigorous Theoretical Analysis.
> Thanks, we agree that our paper focuses on empirical contributions rather than theoretical analysis. While this is the case, our work targets a **highly challenging and practical problem**, cross-domain few-shot object detection, motivated by the limitations of existing approaches, which either require retraining, rely on domain-specific designs, or lack scalability. Facing such a challenging task, we contributed a **novel, effective, training-free, model-agnostic framework** that addresses image generation across domains through the principle of “fix the foreground, adapt the background.” This idea is not only conceptually new but also practically significant.
> Furthermore, we would like to highlight though without mathematical analysis, we extensively validate our method through:
>
> -   **Effective and widely validated** experiments across diverse benchmarks (CD-FSOD, RS-FSOD, Camouflaged FSOD), and across 8 distinct target domains.
> -   **Comprehensive and detailed ablation studies** and analysis demonstrating the contribution of each component.
> -   **Completely training-free**, enabling easy integration into existing detection pipelines without additional overhead.
>
> We believe this combination of **innovative design, strong empirical results, and practical feasibility** also provides valuable insights and a robust solution for the community, complementing theory-driven research.
>
> # W2: Dependence on Quantity of Retrieval DB and Generalization.
>
> We acknowledge the concern and would like to clarify that the domains we tested already include **drastically different scenarios**, such as fine-grained insects (ArTaxOr), cartoon-style drawings (Clipart1k), aerial remote-sensing imagery (DIOR, NWPUVHR-10), underwater cases (UODD, DeepFISH), and industrial textures (e.g., NEU-DET), which are far from the natural image distribution of COCO. Despite this large gap, our method consistently outperforms strong baselines, other competitors, demonstrating robustness even when the retrieval database and target domain differ significantly.
>
> Furthermore, upon question, we conducted additional experiments under three conditions: 1) COCO as DB but with reduced category numbers, 2) NEU-DET (non-general domain) as DB, and 3) also miniImageNet as DB (originally reported in Tab.6, Appendix)
>
> From the below results, we show that: 1) more broader category coverage improves performance, but even with only 1 COCO category, we still clearly enhance the results.  2) Even when the source domain (e.g., NEU-DET) is far from COCO, Domain-RAG still delivers strong gains through its retrieval-conditioned generation and style adaptation modules, 3) As discussed in L927-L935, mini-ImageNet also works as DB, which is also easy to obtain.
>
> From these results, we could conclude that our Domain-RAG enhances the baseline performance regardless of the database, solidly validate our effectiveness.  And, the selection of more general-purpose domains e.g., COCO and ImageNet are widely available, thus require no special design, making our approach **practical and scalable**.
> | Method              | DB                   | ArTaxOr | Clipart1k | DIOR | FISH | NEU-DET | UODD | Avg   |
> |:--------------------|:---------------------|--------:|----------:|-----:|-----:|--------:|-----:|------:|
> | Base GroundingDINO   | -                    |    26.3 |      55.3 | 14.8 | 36.4 |     9.3 | 15.9 |  26.3 |
> | Domain-RAG          | COCO-1classes        |    50.1 |      55.0 | 15.7 | 36.6 |    12.0 | 16.0 |  30.9 |
> | Domain-RAG          | COCO-5classes        |    51.0 |      55.1 | 16.6 | 36.7 |    11.9 | 17.5 |  31.5 |
> | Domain-RAG          | COCO-20classes       |    53.0 |      56.2 | 16.2 | 37.0 |    12.2 | 18.9 |  32.3 |
> | Domain-RAG          | **COCO-80classes (Ours)**|    **57.2** |      **56.1** | **18.0** | **38.0** |    **12.1** | **20.2** |  **33.6** |
> | Domain-RAG          | NEU-DET              |    49.8 |      55.2 | 16.4 | 37.0 |    12.0 | 16.1 |  31.1 |
> | Domain-RAG          | miniImageNet (Tab.6) |    55.6 |      53.2 | 15.6 | 38.0 |    14.0 | 16.2 |  32.1 |
>
> # Q1: Additional Controls or Safeguards to Mitigating the Foreground Leakage?
> Thanks, we appreciate the discussion. We also considered this question ourselves during the method development and Reviewer-wq5x (Q3) also asked about this.  Here are our responses:
>
> Both we authors and Reviewer-wq5x consider adding post-filtering as a safeguard; however, it is non-trivial to achieve a good and especially efficient filtering strategy. Specifically,
>
> 1) Directly applying ZSL inference of existing detectors for foreground detection on CDFSOD is weak, making it difficult for a pseudo-detector to accurately identify leakage. While if we want to apply the trained detector, e.g., trained Domain-RAG (based on GroundingDINO), it will largely burden the pipeline to: data augmentation → train model → filter → train model again.
>
> 2) We also consider applying VLMs (e.g., GPT-4o) to help with filtering. However, they also struggle to meet our expectations.   In the meantime, it will also increase the time cost and complexity.
>
> Besides, we analyzed leakage cases and found that among all generated images in the CDFSL, for example,  only very few examples exhibited a foreground leakage issue. For example, 1.3% (4/290) under 1shot.
>
> Considering these trade-offs, and given the extremely low frequency of leakage (1.3%), we decided not to introduce post-filtering as it would not significantly impact overall performance while violating the method’s training-free and lightweight design.
>
> # Q2: Adapt to Zero-Shot Scenarios?
>
> Thanks. Our method assumes access to target-domain foreground samples, which are provided in any few-shot tasks. Fully zero-shot scenarios, where no target images are available, fall outside this scope.  We will better clarify in the revised version.
>
> # Q3: Selection of Hyperparameter (λ₁, λ₂).
>
> The 1-shot results of different blending coefficients (λ₁, λ₂) are provided below. Experiments show that λ₁ = 1 and λ₂ = 0.8 achieve the best balance. When λ₂ = 1, we find that  λ₁ = 1 performs best. During testing, we slightly reduced λ₂ to 0.8 because setting it too low (e.g., λ₂ = 0.5) distorts semantics and degrades generation quality. This design ensures that domain-specific cues are preserved while still incorporating context from the retrieval database.
>
> | Hyperparameters (λ₁, λ₂) | (0.5,1) | (0.8,1) | (1,1)  | **(1,0.8)** | (1,0.5) |
> |:-------------------------|--------:|--------:|-------:|:-----------:|--------:|
> | **ArTaxOr Performance**  |   51.9  |   52.3  |  53.1  |  **57.2**   |   54.7  |
>
> # Q4: Caption of Figure 3 contains a typo. (a) → a).
> Thanks for your careful review. We will fix it!
>
> # Q5: Effect of Augmented Sample Count per Category.
> Thanks, we conducted the ablation of different numbers of augmented samples per category (n) in Tab.8, Appendix (where “m” was incorrectly written instead of “n”; this will be corrected).  n is selected as 5 according to the ablation results. Note that once we select the best n, we keep it the same for all 3 tasks and 8 targets, without manually tuning it on each target.
>
> We hope that our response adequately addresses your concerns. We are happy to provide more information during the discussion phase.

---

### Official Review · Reviewer_wq5x · 2025-07-03

**Clarity:** 3
**Significance:** 3
**Originality:** 3
**Rating:** 4
**Confidence:** 4

**Summary:**

This paper addresses the problem of Cross-Domain Few-Shot Object Detection (CD-FSOD), where the goal is to detect previously unseen objects in a new domain using only a handful of labeled examples. The authors introduce Domain-RAG, a retrieval-guided, training-free image compositional generation framework that generates domain-aligned synthetic training samples by preserving the foreground and synthesizing new, stylistically and semantically matched backgrounds. The paper details the method's three key components: domain-aware background retrieval, domain-guided background generation, and foreground-background composition on multiple cross-domain FSOD tasks, setting new state-of-the-art results across CD-FSOD, remote sensing FSOD, and camouflaged FSOD benchmarks.

**Questions:**

1: How does database bias or incompleteness in the retrieval gallery affect the robustness and effectiveness of Domain-RAG? Have the authors considered using or simulating more limited/biased galleries or out-of-distribution domains?

2: Have the hyperparameters for retrieval (m, n) and blending coefficients ($\lambda_1$, $\lambda_2$) been systematically explored for different domains or settings? Is the method robust to these values, or does it need careful tuning?

3: Could a post-filtering step be introduced to address foreground leakage/failure cases (e.g., a simple trained foreground/background discriminator, or detection model–based filtering)?

**Ethical Concerns:**

["NO or VERY MINOR ethics concerns only"]

**Final Justification:**

After considering the rebuttal, most of my concerns have been addressed. I decide to raise my rating to borderline accept.

**Limitations:**

Yes.

**Paper Formatting Concerns:**

N/A.

**Quality:**

3

**Strengths And Weaknesses:**

Strengths:

1: The proposed Domain-RAG does not require any model retraining or extra supervision, making it easily adaptable and complementary to various detectors without end-to-end pipeline changes.

2: The method is motivated by shortcomings of existing data augmentation and generation methods for CD-FSOD, and builds on retrieval-augmented generation, a principled approach in the context of domain shift.

3: The approach is extensively validated on three difficult tasks (CD-FSOD, RS-FSOD, camouflaged FSOD) and a variety of domains. Table 1 demonstrates consistent improvements over strong baselines in all evaluated settings. Meanwhile, the paper also provides a thorough ablation analysis (see Figure 3) that critically assesses the contribution of each pipeline component, and visual results (Figure 3b) that illustrate successful compositional alignment and realistic image generation. These concrete results substantiate key claims on both performance and qualitative output.

4: The authors explicitly acknowledge method limitations and provide qualitative diagnostic insights, such as the foreground leakage failure mode in Figure 3c.


Weaknesses:

1: **Lack of Ablation on Synthetic Sample Quantity n**. While the core contribution of the paper lies in leveraging generative models for training-free data augmentation, the paper does not provide any ablation study on the number of generated images (i.e., the hyperparameter n). Given that the number of synthesized images directly affects the support set size and potentially the detector's performance, it is important to understand how sensitive the method is to this parameter. Without such analysis, it is difficult to assess the scalability or robustness of the proposed augmentation strategy.

2: **Fairness and Generalizability in Experimental Comparisons**. From my understanding, the proposed Domain-RAG method is detector-agnostic, yet all the main results are reported only on top of GroundingDINO. It remains unclear whether the compared methods (e.g., CD-ViTO, ETS, Detic-FT) were also allowed to use additional synthetic data during training, or were constrained to the original support set only. This raises concerns about the fairness of the comparison. Furthermore, no experiments are provided to test the generalizability of Domain-RAG when applied to other object detectors.

3: The success of the retrieval component is contingent on the **diversity of the background image database (COCO in this work)**. The paper does not thoroughly analyze or quantify the effect of database bias or incompleteness on augmentation quality, which is critical for practical deployment in truly novel domains or non-natural image settings. A discussion on this is missing from Section 4.1, and the experiments focus on domains where retrieval from COCO is plausible.

4: **Analysis on Performance Drop at Higher Shot Settings.** In Section 4.1 and Table 2, the method shows a slight drop in mAP at higher shot counts (e.g., 20-shot in RS-FSOD). The paper mentions that this might be caused by overfitting or too much synthetic data, but this is only a guess. There is no detailed analysis or experiment to confirm the reason or show how to fix it.

---

> ### Author Rebuttal · Authors · 2025-07-31
>
> Thank you very much for the positive and constructive feedback! We really appreciate that you consider our work “easily adaptable”, “motivated by shortcomings of existing methods”, “extensively validated”, and with “ realistic image generation” . Below we address the concerns raised.
>
> # W1: Ablation on Synthetic Sample Quantity n.
> Thanks for pointing this out. We actually provided an ablation study on the number of generated samples in Tab.8, Appendix (where “m” was incorrectly written instead of “n”; this will be corrected). Results show that performance improves significantly as n increases from 1 to 3, achieves best beyond n=5, and drops when n is set as 10. As in L941-950, we analysis that too few retrieved images limit visual variation, while too many increase the chance of unrealistic compositions.
>
> # W2:  Fairness and Generalizability in Experimental Comparisons.
> Thanks, we first clarify the reason for building our model on GroundingDINO, discuss more on the fairness, and then provide more results regarding the generalizability.
>
> -   **Reason for building upon GroundingDINO.** The primary reason we report results on GroundingDINO is due to its superior performance compared to other detectors (GroundingDINO, 26.3, 1shot, Tab.1) vs. (CD-ViTO, 13.9, 1shot, Tab.1)  of average mAP,  which we believe will advance this area rapidly.
> -   **Fairness of Comparison Regarding the Usage of Synthetic Data.** First of all, though we use COCO as database to augment new images, COCO itself is defined as the source in the default CD-FSOD setting, we don’t break this formulation. 2) The comparison between ours (33.6, Tab.1) and other competitors e.g., ETS (27.8, Tab.1),  Copy-Paste (27.9, Tab.4), Foreground Aug. (28.1, Tab.4), and Background Aug. (29.5, Tab.4) are fair: we share the exact same baseline and training configuration, and all of them are data augmentation based approaches. For example, ETS proposes a mixture of 6 various augmentation strategies during training; The comparison between ours and CD-ViTO and Detic-FT is unfair, but we clearly notice that we use different baselines in Tab.1, and we are not using these performance gap to show our effectiveness.  3) Additionally, the ablation experiments on Fig.3(a) and Tab.5 (Appendix) all use COCO as DB while without applying the complete Domain-RAG method. We believe these results confirms that the improvements stem from our pipeline design rather than merely the introduction of synthetic data.
> -   **Generalizability of Domain-RAG to Other Detectors.** In addition to building on top of GroundingDINO, we also demonstrated our results on SAE-FSDet, the prior SOTA method on remote sensing FSOD, as in Tab.2.  Upon question, we further tested Domain-RAG on other detectors on DeViT-FT and CD-ViTO.  Below results demonstrate that Domain-RAG is detector-agnostic and can be seamlessly integrated into different architectures without requiring retraining or architectural modifications.
>
> | Method                     | 3-shot | 5-shot | 10-shot | 20-shot                          | Avg   |
> | :------------------------- | -----: | -----: | ------: | :------------------------------- | ----: |
> | SAE-FSDet (Tab.2)          |  57.96 |  59.40 |   71.02 | 85.08                            | 68.36 |
> | SAE-FSDet + Domain-RAG (Tab.2)|  59.99 |  65.78 |   72.87 | 84.05      | 70.67 |
>
> | Method                      | ArTaxOr | Clipart1k | DIOR | FISH | NEU-DET | UODD | Avg  |
> | :-------------------------- | ------: | --------: | ---: | ---: | ------: | ---: | ---: |
> | DeViT-FT                    |    10.5 |      13.0 | 14.7 | 19.3 |     0.6 |  2.4 | 10.1 |
> | DeViT-FT + Domain-RAG       |    11.5 |      17.0 | 15.6 | 19.1 |     2.4 |  3.0 | 11.4 |
> | CD-ViTO                     |    21.0 |      17.7 | 11.7 | 20.3 |     3.6 |  3.1 | 12.9 |
> | CD-ViTO + Domain-RAG        |    22.5 |      20.1 | 15.7 | 22.4 |     3.6 |  3.3 | 14.6 |
>
> # W3 & Q1:  Reliance on the Diversity of the Background Database (COCO).
> Thanks, we appreciate this discussion. We would like to clarify that the tested domains are far from being COCO-like. They already involve extremely challenging scenarios with significant category shifts (e.g., novel classes unseen in COCO), severe domain shifts (e.g., clipart-style images, aerial imagery), and also non-natural settings (e.g., industrial textures in NEU-DET).
>
> To further validate the impact of database diversity and bias, we conducted additional experiments under three conditions: 1) COCO as DB but with reduced category numbers, 2) NEU-DET (non-general domain) as DB, and 3) also miniImageNet as DB (originally reported in Tab.6, Appendix)
>
> From the below results, we show that: 1) more broader category coverage improves performance, but even with only 1 COCO category, we still clearly enhance the results.  2) Even when the source domain (e.g., NEU-DET) is far from COCO, Domain-RAG still delivers strong gains through its retrieval-conditioned generation and style adaptation modules, 3) As discussed in L927-L935, mini-ImageNet also works as DB, which is also easy to obtain.
>
> From these results, we could conclude that our Domain-RAG improves the base groundingDINO regardless of the database, solidly validate our effectiveness.
>
> | Method              | DB                   | ArTaxOr | Clipart1k | DIOR | FISH | NEU-DET | UODD | Avg   |
> |:--------------------|:---------------------|--------:|----------:|-----:|-----:|--------:|-----:|------:|
> | Base GroundingDINO   | -                    |    26.3 |      55.3 | 14.8 | 36.4 |     9.3 | 15.9 |  26.3 |
> | Domain-RAG          | COCO-1classes        |    50.1 |      55.0 | 15.7 | 36.6 |    12.0 | 16.0 |  30.9 |
> | Domain-RAG          | COCO-5classes        |    51.0 |      55.1 | 16.6 | 36.7 |    11.9 | 17.5 |  31.5 |
> | Domain-RAG          | COCO-20classes       |    53.0 |      56.2 | 16.2 | 37.0 |    12.2 | 18.9 |  32.3 |
> | Domain-RAG          | **COCO-80classes (Ours)**|    **57.2** |      **56.1** | **18.0** | **38.0** |    **12.1** | **20.2** |  **33.6** |
> | Domain-RAG          | NEU-DET              |    49.8 |      55.2 | 16.4 | 37.0 |    12.0 | 16.1 |  31.1 |
> | Domain-RAG          | miniImageNet (Tab.6) |    55.6 |      53.2 | 15.6 | 38.0 |    14.0 | 16.2 |  32.1 |
>
>
> # W3: Analysis on Performance Drop at Higher Shot Settings.
> Thanks for pointing this out.  We conducted further analyze to understand the slight mAP drop observed at higher shot counts (e.g., 20-shot in RS-FSOD, Tab.2). Specifically, we varied the number of generated samples (n) under the 20-shot setting in RS-FSOD. Results on VHR-10 are as below.  Results validate that **reducing n alleviates performance degradation** at high shot counts (e.g., 85.48% with n=1 vs. 84.15% with n=3), which better validate our guess in L256 to L257.
> We highlight that: 1)  This effect is **limited to high-shot scenarios** and can be addressed with a simple adjustment. 2) Domain-RAG only drops in this single case without manually tuning hyperparameter according to the targets,  and achieves **consistent improvements in 1-, 5-, and 10-shot settings** across 3 tasks and 8 distinct target domains.
>
> | Datasets | baseline | NUM=1  | NUM=3  | NUM=5  |
> | :------- | -------: | -----: | -----: | -----: |
> | VHR-10   |    85.08 | **85.48**  | 84.15  | 84.05  |
>
> # Q2: Hyperparameters: Retrieval (m, n), blending coefficients (λ₁, λ₂).
> Thanks, 1) the results of different n are answered in W1, which was provided in Tab.8, Appendix;  2) m is set as 100 (L215),  the model is robust to m since as long as it covers n (e.g.,5) images it doesn’t affect the results in principle; 3) the 1shot results of different blending coefficients (λ₁, λ₂) are provided below.  Experiments show that λ₁ = 1 and λ₂ = 0.8 achieve the best balance. When λ₂ = 1, we find λ₁ = 1 performs best. During testing, we slightly reduced λ₂ to 0.8 because setting it too low (e.g., λ₂ = 0.5) distorts semantics and degrades generation quality. This design ensures that domain-specific cues are preserved while still incorporating context from the retrieval database.
>
> | Hyperparameters (λ₁, λ₂) | (0.5,1) | (0.8,1) | (1,1)  | **(1,0.8)** | (1,0.5) |
> |:-------------------------|--------:|--------:|-------:|:-----------:|--------:|
> | **ArTaxOr Performance**  |   51.9  |   52.3  |  53.1  |  **57.2**   |   54.7  |
>
> We will add these results to the Appendix.
>
> # Q3:  Could Post-Filtering be Introduced to Address Foreground Leakage/Failure Cases?
> Thanks for the suggestion. We also considered this ourselves, however, we ultimately decided not to adopt it for the following reasons:
>
> 1\) we analyzed leakage cases and found that among all generated images in the CDFSL,  only very few examples exhibited foreground leakage issue. For example, 1.3% (4/290) under 1shot.
>
> 2\) While post-filtering strategies could be applied, they present significant challenges:
>
> -   Directly applying ZSL inference of existing detectors for foreground detection on CDFSOD is weak, making it difficult for a pseudo-detector to accurately identify leakage. While if we want to apply the trained detector, e.g., trained Domain-RAG (based on GroundingDINO), it will largely burden the pipeline to: data augmentation → train model → filter → train model again.
> -   We also consider applying VLMs (e.g., GPT-4o) to help with filtering. However, they also struggle to meet our expectations. In the meantime, it will also increase the time cost and complexity.
>
> Considering these trade-offs, and given the extremely low frequency of leakage (1.3%), we decided not to introduce post-filtering as it would not significantly impact overall performance while violating the method’s training-free and lightweight design.
>
>
> We hope that our response adequately addresses your concerns. We are happy to provide more information during the discussion phase.

---

> > ### Author Response · Authors · 2025-08-04
> >
> > Dear Reviewer,
> >
> > Thank you very much for your thoughtful feedback on our submission. We have carefully addressed your comments in our rebuttal.
> >
> > If there are any remaining questions or concerns, we would be more than happy to clarify them during the discussion phase.
> >
> > We sincerely appreciate your time and efforts in reviewing our work and would greatly value any further input you might have.
> >
> > Best regards, Authors of Submission4884

---

> ### Comment · Reviewer_wq5x · 2025-08-06
>
> Thanks to the authors' reply. Most of my concerns have been addressed. I decide to raise the rating to borderline accept.

---

> > ### Author Response · Authors · 2025-08-06
> > **Thanks for the response!**
> >
> > Dear Reviewer,
> >
> > Thank you for taking the time to read our rebuttal and for your positive feedback. We are very happy to know that our response has addressed your concerns. We truly appreciate your constructive suggestions and the updated evaluation, and we will revise the paper accordingly.
> >
> > Best regards,
> > Authors of Submission #4884

---

### Official Review · Reviewer_1qWt · 2025-07-04

**Clarity:** 4
**Significance:** 3
**Originality:** 3
**Rating:** 4
**Confidence:** 4

**Summary:**

Few-shot object detection using generated image conditioned on RAG DB is a novel and interesting approach from this paper. It resolved the problem about domain different regarding few-shot example from source domain to the target test domain. The proposed Domain-RAG framework is training-free and operates on the principle of "fix the foreground, adapt the background". Benchmark results show a superior performance across multiple datasets particularly in the most few-shot especially 1-shot) settings.

**Questions:**

1. how the foreground object selected?

2. how the cross-domain test setup can make sure the performance once the target test sample is different domain from the RAG DB?

**Ethical Concerns:**

["NO or VERY MINOR ethics concerns only"]

**Limitations:**

Yes

**Quality:**

3

**Strengths And Weaknesses:**

Strengths:

1.	The proposed method is designed well by combing the DB for retrieval augmentation generation manner to find the similar background but masking the foreground and use the searched background as condition for generating new background with similar domain pasting the foreground object as a seamless and natural generation. The proposal is easily to be adapted to the existed cross-domain object detection method.

2.	The proposed method performed well on the few-shot cross-domain object detection benchmark especially the most challenge 1-shot setup. with +7.3 mAP in the CD-FSOD setting.

3.	The paper is organized well and and easy to follow. The methodology is explained clearly, the evaluation setup is designed well and to show the main result clearly supporting the proposed approach.

Weaknesses:

1.	The method relied on the external knowledge from RAG DB, where the quality of the DB and how to create the DB is less of discussion toward how it affected the performance. And how to balance the fairness compared to the other method which does not use the external knowledge is better to discuss.

2.	There is lack of ablation discussion about inpainting with and without RAG DB as condition. For the qualitative and quantity result, it is less of the discussion about the retrieval performance and retrieved image how it looks like, and also less of ablation on the generated sample number $n$.

3. The proposed is a combination of the existed model for the critical module used in the pipeline such as the inpainting model, Flux-Redux model, detection model that less of optimization and novelty on the model side but focus on the entire pipeline, however, the pipeline itself is interesting and still have space to optimized as insight for the community.

---

> ### Author Rebuttal · Authors · 2025-07-31
>
> Thank you very much for the positive and constructive feedback! We really appreciate that you consider our work “well-designed”, “seamless and natural generation”, “easy-to-adapt”, “performed well”, and that our writing “well organized and easy to follow”. Below, we address the concerns raised.
>
> # W1-1: More Discussion on the Quality and Construction of RAG DB.
>
> Thanks. We first clarify the reasons for choosing COCO as DB and then discuss more on how the quantity of DB affects the results.
>
> -   **Reasons for taking COCO as DB.** In the defined (closed-source) CD-FSOD setting as proposed in CD-ViTO[10], COCO serves as the only single source data for training, while the others (ArTaxOr, Clipart1k, DIOR, DeepFish, NEU-DET, UODD) work as unseen targets. Using COCO as the RAG DB: 1) does not introduce extra data beyond the default setting, protecting the fairness of comparison. 2) As in L49, we hope to generate diverse backgrounds for the query image, which COCO fits well with diverse categories and general-domain backgrounds, which can better cover novel domains.
> -   **How Quantity of DB Affects Results.** Upon question, we further conduct more experiments with different DB options, including COCO as DB but with reduced category numbers, NEU-DET (non-general domain) as DB, and also miniImageNet as DB (originally reported in Tab.6, Appendix)
>
> From the results below, we show that: 1) broader category coverage improves performance. 2) More general-domain DB (COCO) works better than
> specific-domain (NEU-DET). 3) As discussed in L927-L935, mini-ImageNet also works as DB, will be inferior to COCO preliminary due to its larger foreground, i.e., smaller background. We highlight that our Domain-RAG improves the base GroundingDINO in all the cases, and solidly validates our effectiveness.
>
> | Method              | DB                   | ArTaxOr | Clipart1k | DIOR | FISH | NEU-DET | UODD | Avg   |
> |:--------------------|:---------------------|--------:|----------:|-----:|-----:|--------:|-----:|------:|
> | Base GroundingDINO   | -                    |    26.3 |      55.3 | 14.8 | 36.4 |     9.3 | 15.9 |  26.3 |
> | Domain-RAG          | COCO-1classes        |    50.1 |      55.0 | 15.7 | 36.6 |    12.0 | 16.0 |  30.9 |
> | Domain-RAG          | COCO-5classes        |    51.0 |      55.1 | 16.6 | 36.7 |    11.9 | 17.5 |  31.5 |
> | Domain-RAG          | COCO-20classes       |    53.0 |      56.2 | 16.2 | 37.0 |    12.2 | 18.9 |  32.3 |
> | Domain-RAG          | **COCO-80classes (Ours)**|    **57.2** |      **56.1** | **18.0** | **38.0** |    **12.1** | **20.2** |  **33.6** |
> | Domain-RAG          | NEU-DET              |    49.8 |      55.2 | 16.4 | 37.0 |    12.0 | 16.1 |  31.1 |
> | Domain-RAG          | miniImageNet (Tab.6) |    55.6 |      53.2 | 15.6 | 38.0 |    14.0 | 16.2 |  32.1 |
>
>
> # W1-2: Fairness of Using RAG DB.
> Thanks, we will clarify that:
>
> 1\) As we mentioned in the reason for taking COCO as DB, it is defined as the source in the default CD-FSOD setting; thus, using COCO should be regarded as a different way of leveraging the same source data.
> 2) Several compared methods include ETS (27.8, Tab.1),  Copy-Paste (27.9, Tab.4), Foreground Aug. (28.1, Tab.4), and Background Aug. (29.5, Tab.4), don’t use COCO in a RAG-inspired way, but share the exact same baseline and training configuration with us, and also fall into the data augmentation-based approaches. The performance gap between them and our Domain-RAG (33.6, Tab.1) on 1-shot instead validates the superiority of our RAG-based method, which has not been proposed in this task before. 3) Additionally, the ablation experiments on Fig.3(a) and Tab.5 (Appendix) all use COCO as DB without applying the complete Domain-RAG method. We believe these results confirm that the improvements stem from our pipeline design rather than merely the use of COCO as an external database.
>
>  We will include the new results and discussions in the revised paper.
>
> # W2-1: Ablation on Inpainting with/without RAG DB as Condition.
> Thanks. In Fig. 3(a) and Tab.5 (Appendix), we reported results for w/o Background Retrieval (random COCO background), which shows only limited improvement (31.8 vs Ours: 33.6, 1-shot). To further address your concern, we added experiments for w/o RAG DB (no DB conditioning, where the original image background is used during Domain-Guided Background Generation). We highlight: 1) the comparison results between “w/o RAG DB” vs. “w/o Background Retrieval” or our Domain-RAG indicate that the DB conditioning contributes significantly to final performance,  bringing background diversity into the extremely limited target images. 2) The comparison results between  “w/o Background Retrieval” vs. our Domain-RAG demonstrate that our domain-aware background retrieval stage provides backgrounds that are better aligned with the target domain (as in L308-309, L922).
>
> | Method                          | ArTaxOr | Clipart1k | DIOR | FISH | NEU-DET | UODD | Avg. |
> | :------------------------------ | ------: | --------: | ---: | ---: | ------: | ---: | ---: |
> | Baseline (GroundingDino)        |    26.3 |      55.3 | 14.8 | 36.4 |     9.3 | 15.9 | 26.3 |
> | w/o RAG DB                      |    34.0 |      56.1 | 16.2 | 37.2 |     8.9 | 14.9 | 27.9 |
> | w/o Background Retrieval (Tab.5)|    51.7 |      57.3 | 17.0 | 37.2 |    10.9 | 16.4 | 31.8 |
> | Domain-RAG (Ours)               |    57.2 |      56.1 | 18.0 | 38.0 |    12.1 | 20.2 | 33.6 |
>
> # W2-2: Discussion on Retrieval Performance and Retrieved Image.
> We appreciate this point and thus evaluate the relevance between retrieved samples and its query through human judgment under the 1-shot setting. For most datasets, over 80% of retrieved samples were deemed relevant, except for NEU-DET, where the relevance was lower due to the lack of similar backgrounds in COCO. We also attempted to use GPT-4o for relevance evaluation, but it struggled with context-specific reasoning (e.g., for insect datasets, GPT-4o considered leaves and grass as different contexts, despite both being suitable as natural backgrounds).
>
> | Metric       | ArTaxOr | Clipart1k | DIOR | FISH | NEU-DET | UODD   |
> |:-------------|--------:|----------:|-----:|-----:|--------:|-------:|
> | Irrelevant   |       2 |         5 |   17 |    0 |      14 |      2 |
> | Relevant     |      33 |        95 |   83 |    5 |      21 |     13 |
> | Total-gen    |      35 |       100 |  100 |    5 |      35 |     15 |
> | Percentage| 94.2%|   95% | 83%|100%|  60%| 86.6%|
>
> Furthermore, we 1) provided the examples of the retrieved background in Fig.5 (Appendix, 2nd column), it shows we managed to retrieval domain-consistent, semantic aligned backgrounds, and 2) again the results summarized in W2-1 between “w/o Background Retrieval (random COCO background) (31.8)” and ours (33.6) well validate that our retrieval images perform well for subsequent image generation.
>
> # W2-3:  Less of Ablation on the Generated Sample Number n.
> Thanks, we actually provided an ablation study on the number of generated samples in Tab.8, Appendix (where “m” was incorrectly written instead of “n”; this will be corrected). Results show that performance improves significantly as n increases from 1 to 3, achieves its best beyond n=5, and drops when n is set to 10. As in L941-950, we analyze that too few retrieved images limit visual variation, while too many increase the chance of unrealistic compositions.
>
> ### W3: Potential Optimization of the Pipeline.
> Thanks for recognizing our pipeline as interesting and proposing this suggestion.  While our method builds upon prior diffusion-based inpainting and detection backbones, its key novelty lies in the training-free principle of “fix the foreground, adapt the background” and the carefully integrated pipeline design for addressing the cross-domain image generation. We also considered optimization, for example, applying efficient finetuning, e.g, LoRA, to better fit into targets. However, given extremely few-label examples, e.g,1 shot and also with the domain shift issue, finetuning the model, no matter the number of learnable parameters, doesn’t help improve performance and will burden the current procedure, limiting the quick deployment of our method.  In contact, our model-agnostic, training-free approach provides a practical and broadly applicable solution.
>
> # Q1: Foreground Object Selection.
> Thanks, since we have the object bounding boxes for those support images, i.e., query images to be augmented, we simply fix the objects labeled by those bounding boxes.
>
> # Q2: Performance When the Test Sample is Different from the RAG DB.
> Thanks, we appreciate this question, while we would like to highlight that: 1) our used CD-FSOD benchmark already include the challenging cases where the target domain is far away from the RAG-DB (COCO), covering domains such as fine-grained animals, cartoon-style, remote sensing, underwater, and industry.  For instance, DIOR is a remote sensing image, which is different from the COCO in the visual domain, resolution, camera perspective, and objects. UODD is a more difficult case, containing industry images in gray color. Examples can be found in Fig.3.  2) Even taking NEU-DET, a domain with an extremely large gap with all the others, as the RAG DB, the table below shows we still increase performance over baseline GroundingDINO.
>
> | Method                           | ArTaxOr | Clipart1k | DIOR | FISH | NEU-DET | UODD | Avg. |
> | :------------------------------- | ------: | --------: | ---: | ---: | ------: | ---: | ---: |
> | Baseline (GroundingDINO)         |    26.3 |      55.3 | 14.8 | 36.4 |     9.3 | 15.9 | 26.3 |
> | Domain-RAG (NEU-DET as RAG DB)   |    49.8 |      55.2 | 16.4 | 37.0 |    12.0 | 16.1 | 31.1 |
>
> We hope that our response adequately addresses your concerns. We are happy to provide more information during the discussion phase.

---

> > ### Author Response · Authors · 2025-08-04
> >
> > Dear Reviewer,
> >
> > Thank you very much for your thoughtful feedback on our submission. We have carefully addressed your comments in our rebuttal.
> >
> > If there are any remaining questions or concerns, we would be more than happy to clarify them during the discussion phase.
> >
> > We sincerely appreciate your time and efforts in reviewing our work and would greatly value any further input you might have.
> >
> > Best regards,
> > Authors of Submission4884

---

> > ### Comment · Reviewer_1qWt · 2025-08-08
> >
> > Thanks for the Author's respones, it resolved my concerns about the doamin difference.
> > Hopefully the author could discuss how to create an ideal DB for the performance consideration, this will be a very important practice for utilzing this approch in the real world, I will keep my rating but still positive to this paper.

---

> > > ### Author Response · Authors · 2025-08-08
> > > **Thanks for the response!**
> > >
> > > Dear Reviewer,
> > >
> > > Thank you for taking the time to read our rebuttal and for your positive feedback. We are very glad to hear that our response has resolved your concerns about the domain difference and also you are positive about our submission.
> > >
> > > Regarding your question on creating an ideal database, we agree that this is a very important point for practical applications. We also appreciate your specific comment *(W1-1: More Discussion on the Quality and Construction of RAG DB)*. As shown in the new results provided in W1-1 *(COCO with different numbers of categories used as DB, NEU-DET as DB, and mini-ImageNet as DB)*, we summarize our findings below as considerations for selecting an ideal DB:
> > >
> > > **Rich Semantic Information**: Including a broad range of categories helps diversify background content *(e.g., COCO-80 classes DB > COCO-20 classes DB > COCO-5 classes DB > COCO-1 class DB)*.
> > >
> > > **Broad Domain Coverage**: Databases with general, natural domains perform better for various unseen targets compared to very specific and distinct domains *(e.g.,  COCO DB, mini-ImageNet DB > NEU-DET DB)*.
> > >
> > > **Minimal Distracting Foreground Content**: A larger proportion of background compared to foreground is preferable, as this prevents the generated models from being misled by foreground objects in the DB. Datasets with bounding boxes *(e.g., COCO)* are advantageous because we can apply inpainting to remove the foreground if needed *(e.g.,  COCO DB > mini-ImageNet DB)*.
> > >
> > > We will incorporate these results, detailed discussions, and conclusions into the revised paper to provide clearer guidance for future applications.
> > >
> > > Thank you again for your thorough review and engagement during the rebuttal discussion period. We hope our feedback helps clarify our method and contributes to the community’s understanding.
> > >
> > > Best regards,
> > >
> > > Authors of Submission #4884

---

### Note · Authors · 2025-08-13

We sincerely thank the AC for coordinating the review process and Reviewers (1qWt, wq5x, vndr, Dhsv) for their constructive engagement.

Our initial scores of 4, 3, 4, 4 (three borderline accepts, one borderline reject) improved to **4, 4, 4, 4 (all positive)**. In particular:

1. We addressed R-wq5x’s (score 3 → 4) concerns, leading all four reviewers to agree on a positive recommendation.
2. While R-vndr was absent during discussion, R-1qWt and R-Dhsv (both rated 4) confirm that we **resolved their concerns** and are positive about our work (1qWt).

**Rebuttal outcomes:**

- **R-1qWt:** Main concern on domain differences was resolved with new results showing strong performance across multiple DBs. We also clarified fairness issues, added further results, and welcomed the suggestion to expand discussion on ideal DB construction.
- **R-wq5x:** Score was raised from 3 to 4 after rebuttal showed 1) fair comparisons, 2) generalization to diverse base methods, 3) robustness to different DB choices. Minor issues such as hyperparameters were also addressed.
- **R-vndr:** Absent from discussion. In rebuttal, we emphasized that while our contribution is not theoretical, it solves CD-FSOD with a novel, training-free, model-agnostic framework. New results showed robustness to poor-quality DBs; all other questions were answered in detail.
- **R-Dhsv:** Maintained positive score after rebuttal clarified 1) rationale for using CD-FSOD and that experiments extend beyond it, 2) fairness evidence, 3) model efficiency, and 4) requested ZSL GroundingDINO results plus applicability to DeViT and CD-ViTO.

**Key Contributions:**

- **Novel method:** Domain-RAG for the first time introduces RAG to FSOD image generation, following “fix foreground, adapt background” by retrieving backgrounds to generate style- and semantics-matched images.
- **Consistent and significant improvements:** Builds new SOTA results across 3 tasks (CD-FSOD, Remote Sensing FSOD, Camouflage FSOD) and 8 targets, with large 1-shot gains (e.g. +7.3 mAP on CD-FSOD).
- **Plug-and-play:** Can be applied to existing detectors with no extra training or effort, enhancing models such as GroundingDINO, SAE-FSDet, DeViT, and CD-ViTO.

Given the resolved concerns, improved scores, and consistent positive feedback, along with the method’s novelty, strong empirical results, and practical applicability, we hope the AC will recognize the substantial merit of this work in the final decision.

---

### Decision · Program_Chairs · 2025-09-17

**Decision:**

Accept (poster)

**Comment:**

This paper works on the problem of Cross-Domain Few-Shot Object Detection (CD-FSOD). It introduces Domain-RAG, a retrieval-guided and training-free image compositional generation framework that synthesizes domain-aligned backgrounds that preserve original foreground objects, and new, stylistically and semantically matched backgrounds. Extensive experiments demonstrate that Domain-RAG achieves consistent improvements over existing methods.

The paper received three borderline accepts and one borderline reject initially. The main strengths include: 1) the method is well designed and motivated, and can be adaptable and complementary other methods; 2) the results look promising with extensive evaluation; 3) the paper is well written and easy to follow, and acknowledges the limitations. However, the main weaknesses include: 1) missing some key discussion; 2) missing ablation and comparison; 3) the proposed method is on the top of existing techniques, and the novelty is somehow limited; 4) the paper is of empirical findings without theoretical analysis and insights; 5) the proposed method is general and should be tested in other applications; 6) missing efficiency/speed analysis.

After the rebuttal, almost of all concerns of the reviewers were well addressed, and they remained positive of this paper. The AC agrees to accept this paper. The authors should follow the reviewers' feedback to improve the final version.